# The Vegetation Composition and Carbon Stock of Old Shrub Typology to Support the Rehabilitation Program in Sumatra and Kalimantan Islands, Indonesia

I. Wayan Susi Dharmawan *, Nur Muhammad Heriyanto, Titiek Setyawati, Marfuah Wardani [ID], Adi Susilo, Raden Garsetiasih, Reny Sawitri, Denny, Vivi Yuskianti, Endang Karlina, Mariana Takandjandji, Rozza Tri Kwatrina and Zuraida

National Research and Innovation Agency, Jl. Raya Jakarta Bogor Km. 46, Bogor 16911, West Java, Indonesia
* Correspondence: iway028@brin.go.id

**Abstract:** The typology of certain old shrubs assists with the selection of suitable plant species for rehabilitation. The carbon stock dynamic in old shrubs is fundamental due to their high uptake during the growth process phase. A plot of 100 m × 100 m (1 hectare) was created in each location, referring to the work of Mueller-Dombois and Ellenberg. The plot was further divided into subsquares measuring 20 m × 20 m; in each plot, there were 25 subplots. Research results showed that the diversity index of old shrubs at the tree, pole, and seedling stage is 2.50–2.66, 1.23–1.50, and 0.67–1.11, respectively. For Kalimantan, the diversity index is lower than that on Sumatra Island, which is 1.64–1.80, 1.00–1.02, and 0.52–0.81, respectively. The carbon stock of the old shrub forest in Sumatra has an average of 36.61 Mg C per ha (standard deviation 14.54 Mg C per ha) to 72.50 Mg C per ha (standard deviation 25.61 Mg C per ha), while Kalimantan has an average of 47.94 Mg C per ha (standard deviation 13.30 Mg C per ha) to 144.07 Mg C per ha (standard deviation 54.64 Mg C per ha). The dynamics of the vegetation composition and carbon stock in each old shrub's typology are considered when choosing a suitable model, including high carbon stock content, to provide optimal results for rehabilitation activities.

**Keywords:** dynamics; diversity index; tree; pole; seedling

## 1. Introduction

Lowland tropical rainforests are generally species rich, especially in primary forests. In general, lowland primary forests in Sumatra and Kalimantan are dominated by plant species from the Dipterocarpaceae family. Primary forest ecosystems can transform into secondary forests and old or young shrubs due to various forest disturbances, including fires, natural disasters, and non-environmentally friendly logging. The old shrub is a small- to large-sized perennial woody plant dominated by small perennial woody plants [1]. Unlike herbaceous plants, old shrubs have persistent woody stems aboveground. Shrubs may be deciduous or evergreen. The old shrub has the advantage of greater tolerance for mechanical disturbances, and they are usually outcompeted by the surrounding trees in terms of light acquisition. In the forest, they are more frequently found in early successional stages [2].

The size of the old shrub area in Indonesia's dryland is 11,330,200 ha, whereas in Sumatra Island, it covers 2,941,600 ha (the Riau, Jambi, and South Sumatra Provinces comprise 38,700 ha, 775,800 ha, and 275,700 ha, respectively). The shrub area on Kalimantan Island amounts to 4,145,600 ha (that of West Kalimantan and East and North Kalimantan amounts to 334,600 ha and 2,237,200 ha, respectively) [1]. Most of the old shrub areas on the islands of Sumatra and Kalimantan have degraded, primarily as a result of human activities. The rehabilitation program represents the government's efforts to improve land quality and cover. The level of land degradation is strongly influenced by forest land management,

and the land degradation level will affect the vegetation composition [3]. Apart from the forest land management aspect, currently, the phenomenon of climate change also affects the condition of vegetation in various regions worldwide. There have been indications of a decline in plant species richness due to global warming, and different plant lifeforms exhibit varying sensitivities to climate change [4,5]. Temperature and precipitation can affect species richness [4], further influencing ecosystem productivity [6].

Rehabilitation activity requires adaptive vegetation species that are tolerant to specific environmental conditions. Rehabilitation is also related to species compositions, structures, or processes leading to a degraded ecosystem [7]. Using an inappropriate vegetation species for rehabilitation will not provide the maximum results for their growth [8]. Therefore, it is necessary to provide accurate data and information about the vegetation composition from many locations on the islands of Sumatra and Kalimantan. The old shrubs in Sumatra and Kalimantan are high-value vegetation, since they are part of the dry lowland rainforest typical of the Sundaland region, with specific species compositions, as well as diversity [9–12]. However, few studies have explored the selection of potential species for rehabilitation, and little is known about their potential carbon stocks. Determining the typology of old shrubs is crucial for the rehabilitation of degraded areas, especially in relation to the selection of appropriate plant species. In addition, the carbon stock dynamics in old shrubs is fundamental because of their high uptake during the growth process phase. For this reason, the role of old shrubs as $CO_2$ absorbers must be appropriately managed. Carbon dioxide uptake is closely related to standing biomass [13]. The amount of biomass in an area is obtained from the production of biomass density and tree species [14]. Great potential for reducing $CO_2$ levels can be achieved through good forest land management practices [15], one of which can be conducted via the rehabilitation of degraded old shrubs.

Accurate data and information about the vegetation composition from many locations in Indonesia, including in the islands of Sumatra and Kalimantan, is essential to support suitable plant species for the rehabilitation of degraded areas. This study aims to (a) identify the vegetation species and carbon stock dynamics as the basis to determine the typology of old shrubs on the islands of Sumatra and Kalimantan and (b) to assess the typology of old shrubs to support the rehabilitation program and its strategy.

## 2. Materials and Methods

### 2.1. Description of the Study Area and Its Context

The study was conducted in lowland tropical rainforests, especially at the old shrub level, at two locations in the year 2019. These locations were Sumatra Island (the Riau Province and the Jambi Province) and Kalimantan Island (the West Kalimantan Province and the East Kalimantan Province). The old shrub is a small- to large-sized perennial woody plant dominated by small perennial woody plants. The research locations consisted of four plot locations: (a) a one-hectare plot in the Riau Province (0°43′50,50″ N and 101°20′58,80″ E); (b) a one-hectare plot in the Jambi Province (1°27′34,092″ S and 103°13′1,2″ E); (c) a one-hectare plot in the West Kalimantan Province (0°11′30,30″ N and 111°18′59,10″ E); and (d) a one-hectare plot in the East Kalimantan Province (0°33′42,92″ N and 117°27′00,94″ E). These sites are highlighted in Figure 1.

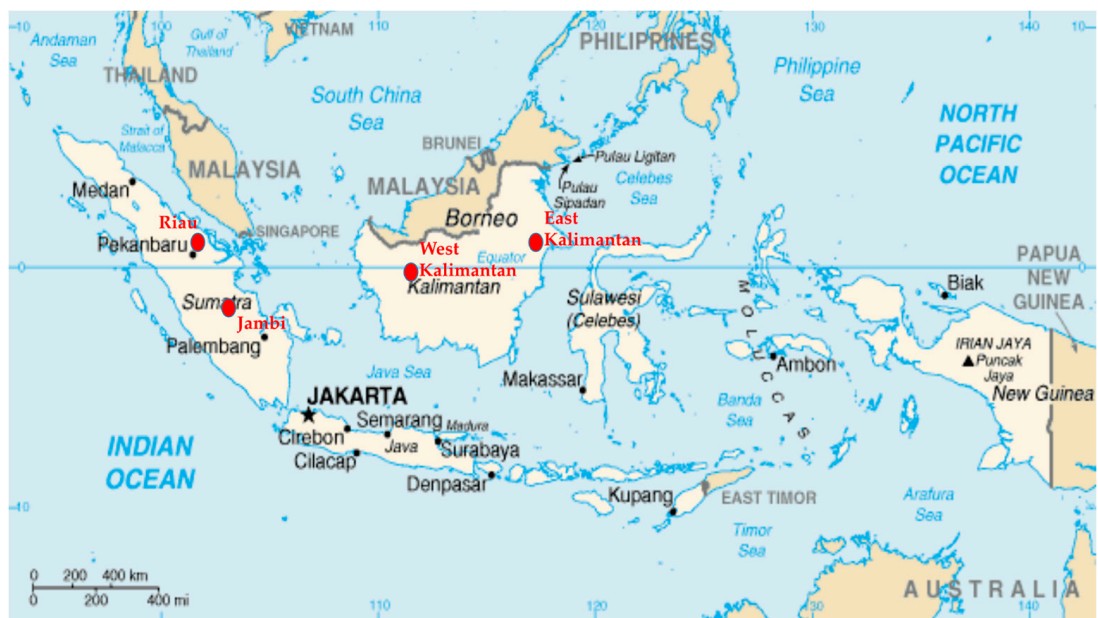

**Figure 1.** The geographic location of the study area in the Riau, Jambi, West Kalimantan, and East Kalimantan Provinces. The Riau and Jambi Provinces belong to Sumatera Island, while West Kalimantan and East Kalimantan belong to Kalimantan Island.

The selection of research locations is based on the representation of vegetation conditions in the study area. The dry lowland tropical rainforests in Sumatra are mainly spread over the Riau and Jambi provinces, with an area of 38,700 ha and 775,800 ha, respectively. In Kalimantan, the dry lowland tropical rainforests are found in the provinces of West Kalimantan and East Kalimantan, covering an area of 334,600 ha and 2,237,200 ha, respectively. The study location was originally a primary natural forest, and later, it was managed and utilized through a selective cutting scheme, becoming a production forest. In subsequent developments, the vegetation progressed into shrubs with the status of a conservation forest.

All of the plots represent the Dipterocarpaceae species. We choose one plot as a representative area of each province, using a purposive sampling approach. The biophysical characteristics of each study site are presented in Table 1.

**Table 1.** Biophysical characteristic of each study site [16].

| Characteristic | Riau Province | Jambi Province | West Kalimantan Province | East Kalimantan Province |
| --- | --- | --- | --- | --- |
| Altitude (asl) | 60 | 40 | 60 | 70 |
| Topography (%) | 0–8 (gentle) | 0–3 (flat) | 0–8 (gentle) | 0–8 (gentle) |
| Soil | <ul><li>Red-Yellow Podzolic</li><li>pH 4–6</li><li>Low fertility</li></ul> | Alluvial-Podsolic, Red-Yellow Podsolic Association Reddish Brown | <ul><li>Red-yellow/ultisol podzolic</li><li>pH 4–6</li><li>Low fertility</li></ul> | <ul><li>Podsolic Red-Yellow/Ultisol</li><li>pH of 4–6</li><li>Low fertility</li></ul> |
| Vegetation category | Pamah tropical rainforest | Lowland tropical rainforest | Lowland tropical rainforest | Lowland tropical rainforest |

### 2.2. Survey and Sampling Designs

A plot of 100 m × 100 m (1 hectare) was created in each location, following the techniques outlined in [17]. The plot was divided into subsquares measuring 20 m × 20 m so that each plot contained several subplots (Figure 2). A 5 m × 5 m subplot was created in each 20 m × 20 m area, and a 2 m × 2 m subplot was created in each 5 m × 5 m

area (Figure 2), respectively. In total, we created 4 plots containing 100 subplots for trees, 100 subplots for saplings, and 100 subplots for seedlings.

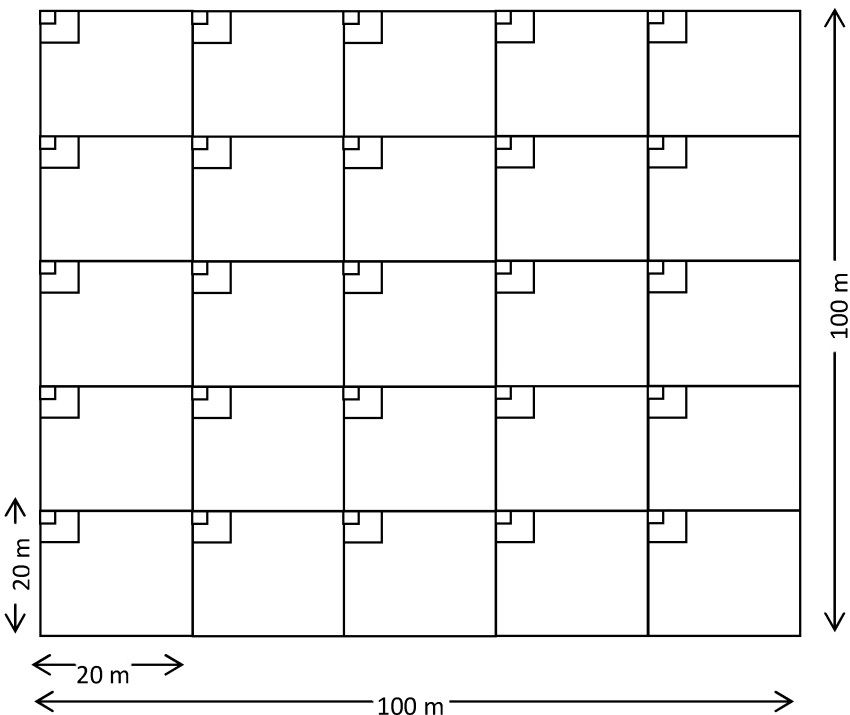

**Figure 2.** Plot design of each study site in Riau, Jambi, West Kalimantan, and East Kalimantan provinces.

The research design adopts purposive random sampling to obtain accurate analytical results. Old shrubs were categorized into three size classes, according to their developmental stage, i.e., (1) seedling—height < 1.5 m (2) sapling—DBH < 10 cm and height ≥ 1.5 m, and (3) tree—DBH ≥ 10 cm. The criteria for tree, sapling, and seedling levels follow those used in vegetation studies [18,19]. Trees were measured in 20 m × 20 m plots. Saplings were measured in subplots of 5 m × 5 m, and seedlings were counted in 2 m × 2 m plots. Measurements of diameter and total height, along with the recording of species names, were carried out at the tree and sapling levels in the subplots. The calculation of the numbers and the recording of the species names were carried out at the seedling level. The vegetation composition and carbon stock were analyzed in terms of the plot locations. For the unknown species, samples of material such as leaves, flowers, and fruits, as herbarium material, were taken to the herbarium at the Center for Forest Research and Development, Bogor, for further identification, as demonstrated in [20].

Soil sampling was conducted in the study site using a composited sample method. Composited samples of soil materials were collected from 5 points in each 100 m × 100 m plot with 30 cm depth, using a ring sampler. The soil samples were analyzed in the Soil Research Laboratory, Ministry of Agriculture, Republic of Indonesia. Chemical characteristics, such as pH, C, N, P, organic matter (OM), CEC (cation exchangeable capacity), BS (base saturation), Ca, Mg, K, Na, Al, and H, were also analyzed.

*2.3. Estimating Vegetation Composition and Diversity*

Changes in the vegetation composition and plant species diversity of the forest ecosystem are commonly used to reflect the impact of any intervention in the system. One of the parameters used to estimate the changes is the use of the important value index (IVI). The IVI is obtained from the sum of the vegetation analyses, which includes the value of relative frequency, relative density, and relative dominance [17,21]. The equation for calculating

relative density, relative frequency, relative dominance, and the important value index is described as follows:

$$\text{Relative density (\%)} = \frac{\text{number of individuals for a species}}{\text{total number of individuals for all species in the plot}} \times 100\% \qquad (1)$$

$$\text{Relative frequency (\%)} = \frac{\text{frequency of a species}}{\text{total number of frequency for all species in the plot}} \times 100\% \qquad (2)$$

$$\text{Relative dominance (\%)} = \frac{\text{total base area of a species}}{\text{total base area of all species in the plot}} \times 100\% \qquad (3)$$

$$\text{Important value index (\%)} = \text{Relative density} + \text{relative frequency} + \text{relative dominance} \qquad (4)$$

Vegetation diversity is calculated using the Shannon diversity index equation, as follows [17]:

$$\text{Shannon's diversity index} = -\sum_{i=1}^{T} \text{abundance index} \times \ln \text{ of abundance index} \qquad (5)$$

Completely randomized design (CRD) and one-way ANOVA at a significance level of 5% were used to detect differences in plant diversity between study sites. Furthermore, the effect was tested to compare the means of all diversity using Duncan's multiple range test (DMRT). SPSS Version 8.0 software was used to support statistical analysis.

*2.4. Estimating Carbon Stock*

Carbon stock estimation was developed based on the above-ground biomass calculated from all the plots established in the study area. The biomass and carbon stocks were evaluated at a tree with DBH > 10 cm. The calculation of stand biomass was analyzed using an allometric equation, as follows [22]:

$$\text{Stand biomass (kg/tree)} = 0.0559 \times \text{wood density} \times \left(\text{diameter at breast height}^2\right) \times \text{height} \qquad (6)$$

where the unit measurement of the diameter at breast height is in cm, the unit measurement of wood density is in $gr/cm^3$, and the unit measurement of height is in m. This allometric equation used the average wood density of $0.61 \, gr/cm^3$. The average rainfall in the research location is between 1500 mm/year–4000 mm/year.

The calculation of the carbon content and the carbon dioxide values was performed using the following equations [21,23]:

$$\text{Carbon stock (kg)} = \text{dry organic matter (kg)} \times 47\% \qquad (7)$$

$$\text{Carbon dioxide (kg)} = \text{Carbon stock (kg)} \times 3.67 \qquad (8)$$

Ultimately, the calculation of the total aboveground tree biomass for each plot was obtained from the sum of all the estimated aboveground biomass values of each tree in the plot and expressed in Mg (mega grams). All of the measured parameters, along with the subsequent analysis, are presented in Table 2.

**Table 2.** Parameters measured and analysis performed.

| Plot | Parameter | Type of Analysis |
|---|---|---|
| • 20 m × 20 m<br>• 5 m × 5 m | Name of species | Species composition<br>Endemicity<br>Conservation status |
| | Number of species | Species density |
| | Frequency of species | Important value index<br>Shannon index |
| | DBH | Stand biomass<br>Carbon stock |
| | Height | Stand biomass |
| | | Carbon stock |
| 2 m × 2 m | Name of species | Species composition<br>Endemicity<br>Conservation status |
| | Number of species | Species density |
| | Frequency of species | Important value index<br>Shannon index |

## 3. Results

### 3.1. Vegetation Composition and Regeneration

A total of 41 species, 33 genera, and 24 families were recorded from the Jaung River old shrub forest area, Sintang, West Kalimantan. The four families with the highest number of species included Rubiaceae (six species), and three families were identified with three species each (Apocynaceae, Clusiaceae, and Euphorbiaceae). *Porterandia glabrifolia* Ridl. experience complete regeneration in the seedling, sapling, and tree stages.

This study recorded 103 species from 79 genera and 38 families in the old-growth secondary protection forest of the Menamang Kanan River, Kutai Kartanegara District, East Kalimantan Province. The four families with the highest number of species were Lauraceae (11 species), Rubiaceae (10 species), Annonaceae (6 species), and Lamiaceae (6 species). The dominant tree species with the highest IVI and a complete regeneration phase was *Melicope lunu-akenda* (Gaertn.) T.G. Hartley.

In the Rantau Bertuah Siak Riau conservation forest area, 51 species of tree (46 genera and 29 families), 32 species of sapling, and 13 species of seedling stages were recorded. *Calophyllum macrocarpum* Hook.f. (IVI = 27.88%), with 74 individuals per ha, *Shorea dasyphylla* Foxw. (IVI = 21.98%), with 40 individuals per ha, and *Gironniera nervosa* Planch (IVI = 19.99%), with 50 individuals per ha showed high dominance based on tree density per hectare. *Calophyllum macrocarpum* Hook.f. is the dominant species, and it undergoes a complete regeneration stage.

In the conservation forest areas of Bangko Lake, Batang Hari, Jambi Province, 61 tree species, 54 genera, 28 families, 33 species of saplings, and 23 species of seedlings were recorded from 25 subplots. The species richness of the tree level in the study sites is far higher than that shown in the results of survey [24], where 54 species were recorded per hectare. However, the number of families in the study site is relatively low compared to the survey results in the old secondary forest (37 families) and the young secondary forest (29 families), as recorded [24] in the Batanghari, Jambi province. *Archidendron bubalinum* (Jack.) Nielsen shows a complete regeneration stage.

The distribution of individual plants in the canopy layer can be interpreted as the distribution of trees per unit area in the various diameter classes. It is referred to as the forest stand structure [25]. The stand structure based on the tree diameter class in the research plot is presented in Figure 3.

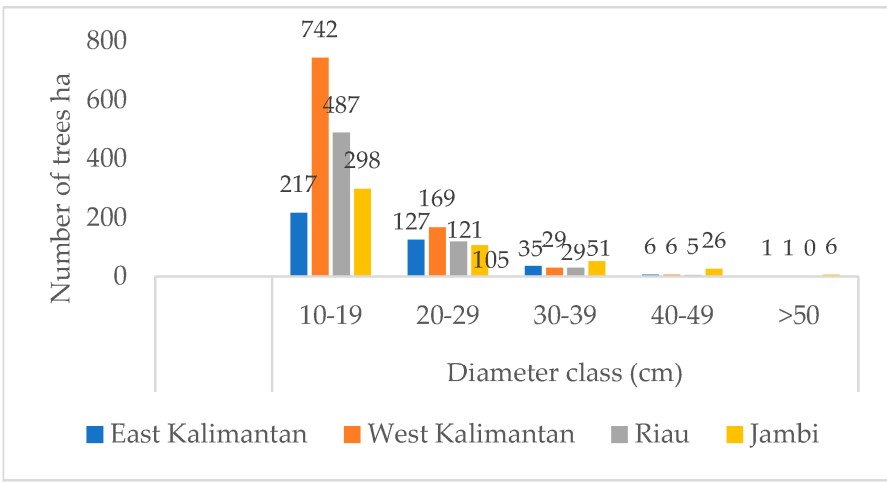

**Figure 3.** Graph of the forest stand structure in the research sites. This graph shows the number of trees per ha in each diameter class (cm) of the research sites.

### 3.2. Carbon Stock Dynamics

The carbon dynamic at the research site varied from 36.61 Mg C per ha to 144.07 Mg C per ha and is strongly influenced by the stand density (N per ha) at each site in Riau, Jambi, West Kalimantan, and East Kalimantan. Overall, the carbon stocks in the study sites in West and East Kalimantan are higher than those in Sumatra (Riau and Jambi) due to the higher stand density at the study sites in Kalimantan compared to those at Sumatra (Table 3). This is also the result of better standing stocks on the study sites of Kalimantan Island, which allow it to absorb and store more carbon. The growth of this stand was influenced by the chemical characteristics of the soil N, P, K, Ca, Mg, Na, and CEC in Kalimantan, which were of better quality than those in Sumatra (Table 4).

**Table 3.** Results of the number of trees per ha, the calculation of biomass and carbon stock, and the estimation of the $CO_2$ equivalent in each research site.

| Site | Number of Trees per ha | Biomass Mg per ha | Carbon Mg per ha | Carbondioxide Equivalent Mg $CO_2$eq per ha |
|---|---|---|---|---|
| Sumatra Island | | | | |
| Riau | Mean = 642 | Mean = 73.22 | Mean = 36.61 | Mean = 134.36 |
| | Median = 625 | Median = 74.47 | Median = 35.00 | Median = 140.00 |
| | Std dev = 200.61 | Std dev = 21.38 | Std dev = 14.54 | Std dev = 46.86 |
| | Std error = 40.12 | Std error = 4.28 | Std error = 2.91 | Std error = 9.37 |
| Jambi | Mean = 486 | Mean = 154.26 | Mean = 72.50 | Mean = 266.08 |
| | Median = 500 | Median = 159.94 | Median = 75.17 | Median = 275.87 |
| | Std dev = 97.92 | Std dev = 54.48 | Std dev = 25.61 | Std dev = 95.79 |
| | Std error = 19.58 | Std error = 10.90 | Std error = 5.12 | Std error = 19.16 |

**Table 3.** *Cont.*

| Site | Number of Trees per ha | Biomass Mg per ha | Carbon Mg per ha | Carbondioxide Equivalent Mg CO₂eq per ha |
|---|---|---|---|---|
| Kalimantan Island | | | | |
| West Kalimantan | Mean = 947 | Mean = 306.54 | Mean = 144.07 | Mean = 528.75 |
| | Median = 1025 | Median = 344.68 | Median = 162.00 | Median = 526.00 |
| | Std dev = 427.78 | Std dev = 116.24 | Std dev = 54.64 | Std dev = 100.77 |
| | Std error = 85.55 | Std error = 23.25 | Std error = 10.93 | Std error = 20.15 |
| East Kalimantan | Mean = 386 | Mean = 102.20 | Mean = 47.94 | Mean = 175.94 |
| | Median = 375 | Median = 103.95 | Median = 48.85 | Median = 179.30 |
| | Std dev = 109.94 | Std dev = 32.03 | Std dev = 13.30 | Std dev = 44.98 |
| | Std error = 21.99 | Std error = 6.41 | Std error = 2.66 | Std error = 8.99 |

**Table 4.** Results of the chemical characteristics at each research site.

| Sites | pH (H₂O) | C-org (%) | N-total (%) | P Bray (I/II) (ppm) | OM (%) | CEC (me 100g⁻¹) | BS (%) |
|---|---|---|---|---|---|---|---|
| Sumatra Island | | | | | | | |
| Riau | 4.3 | 0.41 | 0.06 | 6.9 | 0.71 | 1.42 | 38.82 |
| Jambi | 4.2 | 0.32 | 0.05 | 4.4 | 0.55 | 3.22 | 17.04 |
| Kalimantan Island | | | | | | | |
| East Kalimantan | 4.4 | 0.28 | 0.05 | 1.6 | 0.48 | 3.02 | 15.16 |
| West Kalimantan | 4.2 | 0.48 | 0.09 | 11.0 | 0.83 | 4.93 | 11.03 |

| Sites | NH₄OAc 1N pH 7 Extraction | | | | KCl | |
|---|---|---|---|---|---|---|
| | Ca (me 100g⁻¹) | Mg (me 100g⁻¹) | K (me 100g⁻¹) | Na (me 100g⁻¹) | Al (me 100g⁻¹) | H (me 100g⁻¹) |
| Sumatra Island | | | | | | |
| Riau | 0.17 | 0.04 | 0.07 | 0.26 | 2.01 | 1.29 |
| Jambi | 0.14 | 0.08 | 0.04 | 0.29 | 0.76 | 1.08 |
| Kalimantan Island | | | | | | |
| East Kalimantan | 0.14 | 0.04 | 0.07 | 0.21 | 1.43 | 0.28 |
| West Kalimantan | 0.17 | 0.06 | 0.09 | 0.22 | 3.21 | 1.56 |

The amount of biomass and the carbon content of the forest stands with a diameter of 10 cm is shown in Table 3.

### 3.3. Typology of Old Shrubs on the Islands of Sumatra and Kalimantan

The typology of old shrubs is based on the vegetation composition and the carbon stocks dynamic on the two main islands of Sumatra and Kalimantan, as shown in Table 5. The definition of old shrubs used here is a small- to large-sized perennial woody plant dominated by small perennial woody plants [1], and land-cover vegetation dominated by young trees growing back into the forest. In addition, older forest stands (such as a transitional forest) can still be found [26].

**Table 5.** Old shrubs typology based on the vegetation composition and the carbon dynamic in each research site location. The vegetation composition consists of the dominant plant species and the diversity index. The carbon dynamic consist of the carbon stock and the plant species with a high carbon stock.

| Typology | Sumatra Island | | Kalimantan Island | |
|---|---|---|---|---|
| | **Riau** | **Jambi** | **West Kalimantan** | **East Kalimantan** |
| Vegetation composition | | | | |
| Dominant Species | *Calophyllum macrocarpum* Hook.f., *Shorea leprosula* Miq., *Gironniera nervosa* Planch, *Lithocarpus gracilis* (Korth.) Soepadmo, *Palaquium sumatranum* Burck | *Astronia macrophylla* Blume, *Nauclea orientalis* (L.) L., *Xerospermum noronhianum* (Blume) Blume | *Combretocarpus rotundatus* (Miq.) Danser, *Porterandia glabrifolia* Ridl., *Alstonia scholaris* (L.) R.Br. | *Macaranga gigantea* (Rchb.f. & Zoll.) Mull.Arg., *Melicope denhamii* (Seem.) T.G. Hartley, *Shorea seminis* (de Vriese) Slooten |
| Diversity Index | Mean = 2.66 | Mean = 2.50 | Mean = 1.64 | Mean = 1.80 |
| | Median = 2.63 | Median = 2.43 | Median = 1.56 | Median = 1.88 |
| | Std dev = 0.24 | Std dev = 0.19 | Std dev = 0.33 | Std dev = 0.40 |
| | Std error = 0.05 | Std error = 0.04 | Std error = 0.07 | Std error = 0.08 |
| Carbon Dynamic | | | | |
| Carbon Stock (Mg C per ha) | Mean = 36.61 | Mean = 72.50 | Mean = 144.07 | Mean = 47.94 |
| | Median = 35.00 | Median = 75.17 | Median = 162.00 | Median = 48.85 |
| | Std dev = 14.54 | Std dev = 25.61 | Std dev = 54.64 | Std dev = 13.30 |
| | Std error = 2.91 | Std error = 5.12 | Std error = 10.93 | Std error = 2.66 |
| Species with high carbon stock | *Shorea dasyphylla* Foxw., *Calophyllum macrocarpum* Hook.f., *Lithocarpus gracilis* (Korth.) Soepadmo | *Prunus arborea* (Blume) Kalkman, *Alseodhapne bancana* Miquel, *Agelaea trinervis* (Llanos) Merr. | *Combretocarpus rotundatus* (Miq.) Danser, *Porterandia* sp., *Maccaranga gigantea* (Rchb.f. & Zoll.) Muell.Arg. | *Macaranga gigantea* (Rchb.f. & Zoll.) Mull. Arg., *Melicope lunu-akenda* (Gaertn.) T.G. Hartley, *Callicarpa pentandra* Roxb. |
| Species with endemisity | *Dryobalanops oblongifolia* Dyer. *Intsia palembanica* Miq. | *Euodia aromatica* Blume | *Garcinia dioica* Blume | *Dracontomelon dao* Merr.& Rolfe., *Dipterocarpus caudiferus* Merr. |
| Species with conservation priority | *Intsia palembanica* Miq. | *Lophopetalum javanicum* (Zoll.)Turz. | *Dryobalaops abnormis* V.Sl. | *Eusideroxylon zwageri* Teijsm. & Binn. |
| Biogeography | Flora of Sundaland | Flora of Sundaland | Flora of Sundaland | Flora of Sundaland |

The dominant species in the provinces of Riau, Jambi, West Kalimantan, and East Kalimantan vary according to the characteristics of their respective habitats. This variation is mainly due to the diversity of species at each of these locations, with different diversity indices between locations (Table 6). Based on the results of further tests using Duncan's multiple range test at the tree level, it is indicated that the diversity index at each location on each island does not differ significantly. On the contrary, the locations between islands are significantly different from each other (Table 6). This fact shows that the diversity of species in each site is a vital determinant in the typology of the old shrubs. On the island of Sumatra, the old shrubs diversity index at the tree, sapling, and seedling stages were 2.50–2.66, 1.23–150, and 0.67–1.11, respectively. Meanwhile, on the island of Borneo, the diversity index of old shrubs was lower than that on the island of Sumatra for the tree, sapling, and seedling stages, at 1.64–1.80; 1.00–1.02, and 0.52–0.81, respectively.

**Table 6.** Diversity index at tree, sapling, and seedling stage in each location of the research sites.

| Sites | Diversity Index at Various Plant Stages | | |
|---|---|---|---|
| | **Trees** | **Saplings** | **Seedlings** |
| Sumatra Island | | | |
| Riau | Mean = 2.66 b * | Mean = 1.23 b | Mean = 0.67 b |
| | Median = 2.63 | Median = 1.24 | Median = 0.67 |
| | Std dev = 0.24 | Std dev = 0.28 | Std dev = 0.27 |
| | Std error = 0.05 | Std error = 0.06 | Std error = 0.05 |
| Jambi | Mean = 2.50 b | Mean = 1.50 c | Mean = 1.11 c |
| | Median = 2.43 | Median = 1.49 | Median = 1.09 |
| | Std dev = 0.19 | Std dev = 0.35 | Std dev = 0.43 |
| | Std error = 0.04 | Std error = 0.07 | Std error = 0.09 |
| Kalimantan Island | | | |
| West Kalimantan | Mean = 1.64 a | Mean = 1.00 ab | Mean = 0.52 a |
| | Median = 1.56 | Median = 1.05 | Median = 0.62 |
| | Std dev = 0.33 | Std dev = 0.35 | Std dev = 0.33 |
| | Std error = 0.07 | Std error = 0.07 | Std error = 0.07 |
| East Kalimantan | Mean = 1.80 a | Mean = 1.02 a | Mean = 0.81 b |
| | Median = 1.88 | Median = 1.02 | Median = 0.80 |
| | Std dev = 0.40 | Std dev = 0.41 | Std dev = 0.52 |
| | Std error = 0.08 | Std error = 0.08 | Std error = 0.10 |

* Significant using the one-way ANOVA test at a level of $\alpha < 0.05$. Numbers followed by the same letters are not significantly different.

## 4. Discussion

### 4.1. Vegetation Composition and Regeneration

A study from the lowland rainforest of Kalimantan [27] showed that the floristic composition among locations varies; each has its characteristic genera. This condition is affected by habitat and geographic factors. The species numbers of tree, sapling, and seedling stages in the conservation forest area of Rantau Bertuah Siak, Riau Province, are much lower than those noted in the surveys [24,28]. Meanwhile, the tree species richness in the conservation forest areas of Bangko Lake, Batang Hari, Jambi Province is far higher than the results obtained in other studies [19].

Regeneration is a natural process in the forest ecosystem in which the organism mechanism will renew the forest stand structure to maintain and extend its existence. Young trees will replace mature trees when they have been logged, burned, toppled due to natural disasters, or are physiologically dead. Moreover, these are all natural phenomena. The regeneration process depends on several factors, including the creation of a natural gap in the tropical forest which has undergone a complex process. Vegetation composition and complete regeneration are fundamental in a natural forest ecosystem process in which the organism renews the forest stand structure to maintain and continue its existence. Tree species with complete regeneration are always present at seedling, sapling, and tree stages [29].

The ability of trees to utilize solar energy, nutrients or minerals, and water, as well as the nature of competition, is varied, even in the same sites [30], leading to the variation in diameter class distribution in the forest stand. At this research site, the forest stand structure has been decreasing in terms of the number of trees, from small to large diameter. This is in line with other studies which state that there are more small-diameter classes in natural forests than large-diameter classes [31].

*4.2. Carbon Stock Dynamics*

The carbon dynamic at this site is part of the dynamics of the landscape caused by disturbances to the forested areas, which transform into old shrub forest areas [32]. The growth process in young trees is relatively faster than that in old trees due to the optimization of the photosynthetic process [33].

The biomass and carbon content in the trees at a diameter of 10 cm vary. The results of research [34] in the Central Kalimantan Forest show that the carbon stock can reach 73.08 Mg C per ha, which is higher than the results obtained in this study, except for in West Kalimantan. Meanwhile, the results of research [35] in the secondary forest of Mandiangin, South Kalimantan, estimated a carbon stock of 81.59 Mg C per ha. Forest ecosystems can store carbon and are considered the largest carbon pool in terrestrial ecosystems [36,37]. The great potential for storing carbon in old shrub forests on Sumatra Island (Riau and Jambi) and Kalimantan Island (West Kalimantan and East Kalimantan) is shown in Table 3.

On Sumatra Island (Riau and Jambi), plant species such as *Shorea dasyphylla* Foxw., *Calophyllum macrocarpum* Hook.f., *Lithocarpus gracilis* (Korth.) Soepadmo, *Prunus arborea* (Blume) Kalkman, *Alseodhapne bancana* Miquel, and *Agelaea trinervis* (Llanos) Merr. have great potential for storing carbon in old shrub forests. Meanwhile, on the island of Borneo (West Kalimantan and East Kalimantan), the plant species of *Combretocarpus rotundatus* (Miq.) Danser, *Porterandia* sp., *Maccaranga gigantea* (Rchb.f. & Zoll.) Muell.Arg., *Melicope lunu-akenda* (Gaertn.) T.G. Hartley, and *Callicarpa pentandra* Roxb. have great potential for storing carbon in old shrub forests.

*4.3. Typology of Old Shrubs on the Islands of Sumatra and Kalimantan to Support Rehabilitation in Degraded Forests*

The diversity of species in each site is a crucial determinant for the typology of old shrubs. In this study, the diversity index of old shrubs in Kalimantan was lower than in Sumatra for the tree, sapling, and seedling stages. This study's typology of old shrubs based on vegetation composition and carbon stock dynamics can support proper species selection for degraded forest land rehabilitation.

The dynamic of carbon stock among locations varies, except in West Kalimantan, which has the highest carbon stock of 144.07 Mg C per ha. According to research conducted in Reference [26], the average carbon stock in the old shrubs forest is 60 Mg C per ha. Therefore, the carbon stock in the study sites of Riau, Jambi, and East Kalimantan has a similar value to that estimated in other research [26]. The high carbon stock in the old shrubs forest of West Kalimantan indicated the presence of a transition forest towards the old secondary forest, as shown in Figure 4. Several dominant species in the study sites also become species that can store a large amount of carbon. Examples of such species include *Calophyllum macrocarpum* Hook.f. (in Sumatra Island), *Combretocarpus rotundatus* (Miq.) Danser, *Porterandia glabrifolia* Ridl., and *Macaranga gigantea* (Rchb.f. and Zoll.) Mull.Arg. (on Kalimantan Island).

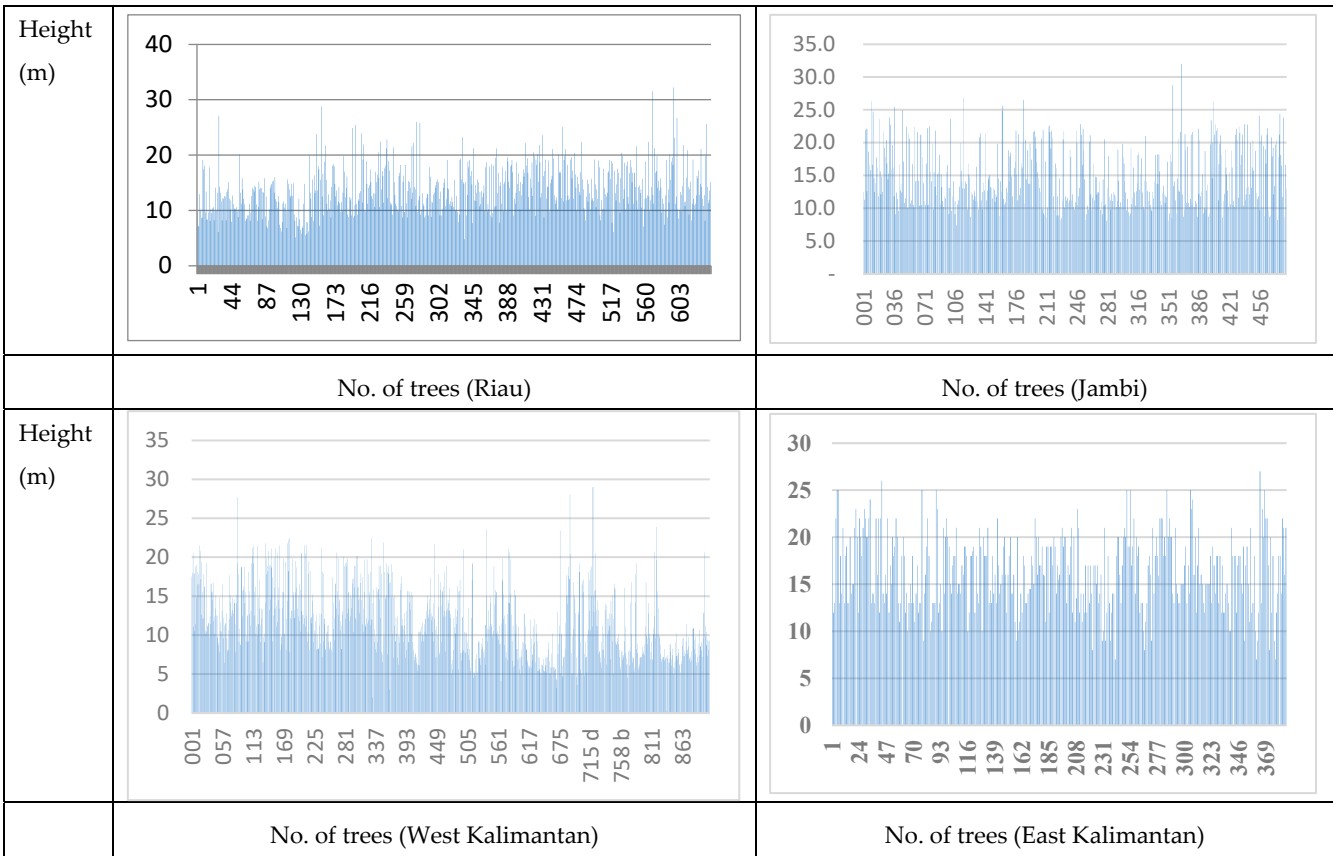

**Figure 4.** Vertical profile of the stands on Sumatra Island (Riau, Jambi) and Kalimantan Island (West Kalimantan and East Kalimantan). This graph shows tree height (m) of each tree.

The selection of plant species primarily determines the success of a rehabilitation program. This study's typology of old shrubs can support the selection of suitable species for rehabilitating degraded forest land. The presence of old shrubs in plant stands indicates previous disturbances caused by logging, fire, or other factors. Unlike herbaceous plants, old shrubs have strong woody stems that grow less than 2 m per year and often form the undergrowth of tall trees. Shrubs are categorized as old shrubs if the forest disturbance has occurred over a long time, while young shrubs are shrubs that have just formed. Thus, if left undisturbed, old shrubs will shortly turn into secondary forests. With plant species that have become old shrubs, a choice of plant species for rehabilitating degraded forest land is available [38]. In addition to the selection of plant species, it is also necessary to examine the growth of these plants, which can be determined based on their biomass and carbon stocks.

In addition, parameters of endemicity characteristics, conservation status, and distribution of biogeographical areas are essential when it comes to selecting valuable species for rehabilitation programs. As represented in Table 3, species that exhibit high endemicity, enjoy protected conservation status, and are native species in the biogeographical distribution area take priority in the development of dry lowland tropical rainforest rehabilitation programs in Sumatra and Kalimantan.

**5. Conclusions**

The old shrubs on Sumatra Island (Riau and Jambi) and Kalimantan Island (West Kalimantan and East Kalimantan) are dominated by endemic and fast-growing species that have the potential to rehabilitate degraded forest land and support environmental services (carbon stocks) ecosystem conservation as a biodiverse habitat. The typology of the old shrubs on Sumatra Island and Kalimantan Island can be determined based on the

vegetation composition and the carbon stocks dynamic. The diversity index of old shrubs at the tree, sapling, and seedling stages on Sumatra Island is higher than that of Kalimantan Island. On the other hand, the carbon stock of old shrubs on Sumatra Island is lower than that of Kalimantan Island.

The dynamics of vegetation composition and carbon stock in each typology of old shrubs can be considered when determining suitable species in order to ensure high carbon stocks to provide maximum results for rehabilitation.

**Author Contributions:** Each author (I.W.S.D., N.M.H., T.S., M.W., A.S., R.G., R.S., D., V.Y., E.K., M.T., R.T.K., and Z.) had an equal role as main contributors who equally discussed the conceptual ideas and the outline, conducted the literature reviews, performed the analysis, prepared the initial draft, provided critical feedback on each section, as well as revised and finalized the manuscript. All authors have read and agreed to the published version of the manuscript.

**Funding:** This research was collaboration between Forest Research & Development Center and Sinar Mas Group.

**Informed Consent Statement:** Not applicable.

**Data Availability Statement:** The datasets used and/or analyzed during the current study are available from the corresponding author on reasonable request.

**Acknowledgments:** We thank the anonymous reviewers for their detailed comments and corrections.

**Conflicts of Interest:** The authors declare no conflict of interest.

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
