# Peer review of "The Vegetation Composition and Carbon Stock of Old Shrub Typology to Support the Rehabilitation Program in Sumatra and Kalimantan Islands, Indonesia"

_sustainability, doi:10.3390/su15021389_

Round 1

Reviewer 1 Report

Dear authors,

I have reviewed your manuscript “The Old Shrub Typology to Support the Rehabilitation Program in Sumatra and Kalimantan Ecoregion Islands, Indonesia”

MS falls within the journal’s scope and the subject matter is quite interesting. However, the Abstract is too lengthy and needs to rewrite precisely. The manuscript mainly deals with the carbon stocks and the typology of old shrubs, but the Introduction failed to motivate and problematize these objectives, so it should be improved. The methodology needs to be revised and the statistical methods/tools/software used for data analysis need to be discussed in detail. In addition, the quality of the Figures should be improved. Moreover, extensive editing of the English language, grammar and style is required, and there are several drawbacks to address. Briefly, the article can't accept in its present form.

The comments and suggestions about MS

Title

Line 2-3: The title should reflect your main results/conclusions. However, the manuscript's title is not suitable. So strongly recommend to modify it.

Abstract

Line 13-29: According to journal (Sustainability) policy, the abstract should not be more than 200 words and your abstract is more than 260. So please rewrite the precise and short Abstract to meet the journal’s requirements and for easy understanding of readers.

Line 30: Please avoid repeating the same words (like old shrub; typology; rehabilitation) that you have already used in the title to enhance the visibility of your article.

Introduction

The authors are encouraged to cite recent literature in the introduction.

https://doi.org/10.1016/j.ecolind.2021.108288, https://doi.org/10.3390/f13010100

Line 34-36: This is an ambiguous sentence, so please rephrase this sentence for a clear understanding of readers.

Line 49-51: This is another ambiguous sentence, so please rephrase this too.

Line 60-61: ‘Carbon dioxide uptake is closely related to standing biomass.’ Reference?  

Materials and Methods

Line 78: Replace ‘These sites are presented Figure 1’ with ‘These sites are presented in Figure 1’.

Line 78-81: A figure/table should be self-explanatory and its caption should be detailed. So, the quality and caption of Figure 1 should be improved.

Line 97-99: Replace ‘A plot of 100 m × 100 m (1 hectare) was made in each location, referring to the techniques. The plot was divided into sub-squares measuring 20 m × 20 m so that one plot contains subplots. (Figure 2).’ with ‘A plot of 100 m × 100 m (1 hectare) was made in each location, referring to the techniques. The plot was divided into sub-squares measuring 20 m × 20 m so that one plot contains subplots (Figure 2).

Line 101-102: How can be the same caption for Figures 1 and 2? Can you justify it?

Line 128-131: Please write in detail the Statistical analyses and mention the software you used for analysis.

Results

Line 180-181: Same comments as for Lines 78-81.

Line 194-195, 204-218: Tables caption should be detailed and mention what type of values you present in the tables, i.e., means, median, standard error, standard deviation, etc.

Discussion

Line 284: The caption should be detailed.

Overall, the discussion is fine.

Conclusions

Conclusions need to be revised as per factual findings and there is no need to write some values in conclusions. Conclusions should be to the point, comprehensive and logical.

In light of the above, I think the MS deserves to be published but should only be accepted after a Major Revision.

Please take into consideration the comments on the PDF file of the revision.

Good luck!

Author Response

Dear Reviewer#1

Best regards,

I Wayan S. Dharmawan

Reviewer 2 Report

I have commented on the attached file to suggest ways to improve the paper.  I was particularly confused by the term 'old shrubs'.  'Shrubs' is traditionally used to describe woody vegetation that grows less than 2 m in height and is often an understory to tall trees. 

Author Response

Dear Reviewer#2

Best regards,

I Wayan S. Dharmawan

Reviewer 3 Report

Dear Authors, 

Please see the following:

Vivi Yuskianti 2 from where is 2?

Pay attention to the reference numbers in the text. All are written with superscripts.

Line 34 – please revise the sentence: ”These conditions are dynamic and stable ecosystems”. The conditions might be characteristic of a stable ecosystem. Also, you talk about species richness only.

Line 34 – 36 – ”The stability of the ecosystem can turn into secondary forest and old or young shrubs due to various forest disturbances, including fires, natural disasters, and not environmentally friendly logging”. Please revise the phrase. Stability is a quality, a state, that cannot turn into a forest or shrub.

 Line 43 – ”whereas Sumatra Island covers 2,941,600 ha” – I suppose is better: whereas in Sumatra Island covers 2,941,600 ha - because it is about shrubs areas not about the island area. Please pay attention to the following sentences also (lines 45-46-47).

Line 57 – please revise the English

Line 62 – ”Great potential in reducing CO2 levels can be done” – my suggestion: can be reached

Line 232 – 233 – ”The regeneration process very much depends on several factors, including natural gap creation in the tropical forest” – the phrase is repeated in lines 236-237

Please revise the English language and you might need to get advice from a senior researcher.

Author Response

Dear Reviewer#3

Best regards,

I Wayan S. Dharmawan

Reviewer 4 Report

The paper entitled The Old Shrub Typology to Support the Rehabilitation Program in Sumatra and Kalimantan Ecoregion Islands, Indonesia is evaluated.

The subject matter is interesting and may prove to be a step forward in knowledge. However, the article presents some important problems in its approach. The methodology, results, discussion and conclusions are not sufficiently related to the parameters analysed: vegetation composition and diversity and estimating carbon stock. In relation to this issue, the research questions are not clear.

The methodology is confusing and the sites where the plots have been established are not well characterised or contrasted. Why was each site chosen? What physical-environmental and, fundamentally, vegetation characteristics define and differentiate them from each other? This deficiency prevents a clear understanding of the work and makes it difficult to follow the storyline of the work.  The section should be completely revised and extended. The methodology should list all the aspects worked on in the greatest detail and in an orderly fashion, explaining what each action is carried out for and what result is intended to be obtained with each method applied and the relationship between all of them. In relation to the results: the information on the differences in the numbers and types of families and genera found in each plot is not sufficiently developed.

Specifically, the following changes are suggested:

Introduction: It is recommended that this section be expanded to include more scientific texts that support the issues addressed in the paper. They can also be taken to the discussion of the results. Several of the research topics are not introduced in this section. 

Line 78.

Figure 1. The map in Figure 1 should have higher resolution. Incorporate graphic or numerical scale. Include the names of the main islands. If possible, include the delimitation of provinces. It is recommended to include topography.

Line 82.

It is recommended to introduce a table that classifies and clarifies the characteristics of each of the analysis locations. It is recommended that each area be assigned a fixed code (letter or number) that identifies them, in order to facilitate the understanding of the results. It would be advisable to characterise each plot from the point of view of its state of vegetation conservation as well as its vegetation evolution.

Line 96. Paragraph 2.2

It is necessary to describe clearly how the data collection is carried out on the established plots, as well as the information that is collected.

Clarify whether species level identification was always achieved and, if not, the percentage of species not identified at this level.

Specify the number of strata identified and the thresholds for their differentiation.

Specify the total number of plots and subplots.

Line 101: change the title of figure 2.

Line 110: it is recommended to include the classification and measurement criteria for tree, sapling, and seedling levels, even if the reference to the methodology is mentioned.

In general, for the methodology section, a table specifying all the parameters that have been measured in each plot as well as the indices obtained per species, plot, etc. is suggested.

Line 200: The definition should be stated in the introduction and methodology.

Line 285: In the last paragraph mention is made of climate change issues that are outside the discussion.

In general, it does not seem sufficient to characterise the formations on the basis of species identification. It would be necessary to include some parameter that informs on the value of each of these species, for example the biogeographical value, possible endemism, etc.

Line 299- 317: these paragraphs are results but cannot be considered conclusions.

Author Response

Dear Reviewer#4

Best regards,

I Wayan S. Dharmawan

Round 2

Reviewer 1 Report

Thanks to the authors for critically addressing my concerns and I am satisfied with the current version.

Reviewer 3 Report

Good work with the changes. 

Reviewer 4 Report

Dear Authors. Thank you for including the suggestions in your latest version. I think the article is much clearer and tidier now. The modifications incorporated by the suggestions of the other reviewers are also interesting.